# A Complex Metabolic Network Confers Immunosuppressive Functions to Myeloid-Derived Suppressor Cells (MDSCs) within the Tumour Microenvironment

**DOI:** 10.3390/cells10102700

**Published:** 2021-10-09

**Authors:** Francesca Hofer, Gianna Di Sario, Chiara Musiu, Silvia Sartoris, Francesco De Sanctis, Stefano Ugel

**Affiliations:** Immunology Section, Department of Medicine, University of Verona, 37134 Verona, Italy; francesca.hofer@univr.it (F.H.); gianna.disario@univr.it (G.D.S.); chiara.musiu@univr.it (C.M.); silvia.sartoris@univr.it (S.S.); francesco.desanctis@univr.it (F.D.S.)

**Keywords:** myeloid-derived suppressor cells (MDSC), cancer, tumour-microenvironment (TME), inflammation, immunometabolism

## Abstract

Myeloid-derived suppressor cells (MDSCs) constitute a plastic and heterogeneous cell population among immune cells within the tumour microenvironment (TME) that support cancer progression and resistance to therapy. During tumour progression, cancer cells modify their metabolism to sustain an increased energy demand to cope with uncontrolled cell proliferation and differentiation. This metabolic reprogramming of cancer establishes competition for nutrients between tumour cells and leukocytes and most importantly, among tumour-infiltrating immune cells. Thus, MDSCs that have emerged as one of the most decisive immune regulators of TME exhibit an increase in glycolysis and fatty acid metabolism and also an upregulation of enzymes that catabolise essential metabolites. This complex metabolic network is not only crucial for MDSC survival and accumulation in the TME but also for enhancing immunosuppressive functions toward immune effectors. In this review, we discuss recent progress in the field of MDSC-associated metabolic pathways that could facilitate therapeutic targeting of these cells during cancer progression.

## 1. Introduction

The tumour microenvironment (TME) is an intricate ecosystem in which different cell subsets coexist, including cancer, stroma, and endothelial cells, and also diverse cell subsets of innate and adaptive immunity. In progressing tumours, both cancer cells and tumour-infiltrating cells are dynamic. Indeed, tumour cells acquire and accumulate different somatic mutations, ultimately resulting in abnormal cellular fitness and growth [1,2]. These genetic modifications influence the composition of the TME to alter tumour immunogenicity and elicit functional and phenotypic changes in both non-immune stromal components and immune cells. Ongoing mutational processes lead to the generation of cancer neoantigens that are potentially recognised by tumour-fighting effector cells such as natural killer (NK) cells, cytotoxic CD8^+^ T, and CD4^+^ Th1 lymphocytes that activate a powerful anti-tumour immune response that is able to achieve substantial tumour debulking [3,4,5]. Conversely, through the oncogenic-driven expression of soluble immune mediators such as chemokines, growth factors, and inflammatory cytokines, transforming cells attract and activate specific components of the immune system that exert suppressive functions, such as regulatory T cells (Tregs), tumour-associated macrophages (TAMs), and myeloid-derived suppressor cells (MDSCs), to promote the establishment of systemic immune tolerance favouring tumour progression [6,7,8,9]. Thus, the characterisation of the cancer genome and the immune landscape of TME are complementary and crucial for predicting disease progression and therapeutic outcome.

Understanding the TME immune context is crucial in the era of cancer immunotherapy [10,11,12]. Indeed, patients with the same cancer type may present a different immune composition within the TME, thus highlighting that the mapping of tumour-infiltrating immune cells and defining their functional state is of paramount importance in terms of diagnosis and the design of therapeutic approaches [11,13]. A broad classification of TME according to the assessment of tumour-infiltrating lymphocytes (TILs) identified two antithetical immune landscapes termed “hot” and “cold” tumours, respectively [14]. Hot tumours are immune-privileged environments that are characterised by an increased infiltration of T cells within the tumour mass. These highly infiltrated tumours are thought to be immunogenic and immunoreactive, as clinical responses to immune checkpoint inhibitors (ICIs) occur frequently in this histological setting [14]. Immunoreactive tumours also display proinflammatory cytokines (i.e., tumour necrosis factor (TNF)α, type I and type II interferon (IFN), and interleukin (IL)-23, IL-22, and IL-1β) that should establish a suitable environment for T cell fitness and activation and possess a higher mutational burden compared to that of other cancer subtypes, and this can act as a source of tumour-associated antigens to prime anti-tumour T cells [15,16,17]. In contrast, immunologically cold tumours resemble two distinct immune landscapes that include immune-excluded and immune-desert TMEs [14]. Both of these cancer settings can be considered to be non-inflamed tumours, and they rarely respond to ICI-based immunotherapy [18]. The paucity of T cells in both the periphery and the parenchyma of the tumour is indicative of TMEs that possess an immune-desert phenotype. These tumours typically reflect the absence of an endogenous anti-tumour T cell response [18,19,20]. In contrast, immune-excluded tumours are characterised by an intense infiltration of immune cells such as myeloid cells and TILs, where the latter cells are restricted to the periphery of the tumour stroma and are thus unable to penetrate the tumour core [18,21]. These peculiar features suggest that immune-excluded cancers may be potentially antigenic but slightly immunogenic. Indeed, in this tumour context anti-tumour TILs are either blocked in the tumour stroma or functionally attenuated. TIL retention is partially dependent upon the presence of a fibrotic barrier that is established by cancer-associated fibroblasts (CAFs) that function to limit T cell trafficking [22]. Recently, a conserved stroma-immune relationship has been identified in 20 different cancers, and this relationship is capable of playing a prognostic role in response to ICI-based immunotherapy [23]. Additionally, alterations in both the adhesion molecules in endothelial cells and the chemokine-based axis can strongly influence T cell homing into the tumour [24,25,26]. The lack of expression of costimulatory receptors and the downregulation of major histocompatibility complex (MHC) molecules on the surface of tumour cells can affect TIL activation [27]. Moreover, incomplete TIL functionality is influenced by the presence of immunosuppressive immune elements within the TME. For example, Tregs abrogate TIL proliferation and survival by competing for the available IL-2 within the microenvironment [28], and this ultimately results in effector T cell apoptosis [29]. Moreover, CD4^+^CD25^+^FoxP3^+^ Tregs upregulate metabolic mediators such as the lipid transporter CD36 to preserve their viability and reinforce their immunosuppressive functions within the TME [30]. Among the immunosuppressive tumour-infiltrating leukocytes, myeloid cells such as MDSCs are predominant and function to guide several aspects of cancer growth [31,32]. By releasing proangiogenic factors, including vascular epidermal growth factor (VEGF) and prokineticin-2 (also known as Bv8), MDSCs support tumour angiogenesis and vasculogenesis [33,34] and promote both stemness and the proliferation of cancer cells [35,36]. Additionally, these factors promote cancer dissemination that can ultimately lead to metastases (as extensively reviewed previously) [37,38,39]. For instance, MDSCs exert suppression of immune effector cells by exploiting four main mechanisms: depletion of essential metabolites, production of reactive oxygen and nitrogen species (ROS and RNS, respectively), as well as immune effector inhibition by either direct contact or release of soluble factors [38]. However, the most proficient pro-tumour function facilitated by MDSCs within the TME is the abrogation of the anti-tumour immune response. To this end, MDSCs exploit several strategies that limit trafficking, fitness, survival, and functions of effector cells by releasing immunosuppressive metabolites, thus establishing a competition for nutrients within leukocytes within the TME and facilitating the expression of inhibitory receptors [40,41].

Here, we present the major metabolic networks that drive the immunosuppressive functions of MDSCs that can shape the TME into an immune-excluded phenotype, and we propose a roadmap for developing effective therapeutic MDSC-targeting approaches to enhance cancer immunotherapy.

## 2. Ontogeny, Phenotype, and Main Characteristics of MDSCs

The MDSC nomenclature is an umbrella definition that encompasses myeloid cells at diverse differentiation stages that express non-exclusive phenotypic markers and share immunoregulatory and pro-tumour functions [42,43,44]. MDSCs were initially identified in tumour-bearing mice based on the co-expression of two antigens, Mac-1 (CD11b) and Gr1 (the anti-mouse Gr1 antibody recognises epitopes present in both Ly6C and Ly6G molecules) [45]. Currently, mouse MDSCs are typically divided into two major subpopulations that include monocytic MDSCs (M-MDSCs, CD11b^+^Gr1^int^Ly6G^-^Ly6C^+^ cells) and polymorphonuclear MDSCs (PMN-MDSCs, CD11b^+^Gr1^high^Ly6G^+^Ly6C^-^ cells) [44,46,47]. Recently, new potential candidates (i.e., CD49b, CD115, CD31) have been proposed, as no definitive and restrictive markers allow for the distinction of M-MDSCs from monocytes and PMN-MDSCs from granulocytes [44]. In the near future, the application of innovative technologies such as fate-mapping analysis and in-depth RNA interrogation at the single-cell level (i.e., scRNA-seq) should resolve the difference between MDSCs and their normal myeloid non-suppressive counterparts. In support of this, a recent scRNA-seq analysis of MDSCs isolated from the spleens and tumours of breast cancer models demonstrated that M-MDSCs and PMN-MDSCs display a gene signature that profoundly differs from that of monocytes and granulocytes, respectively [48]. Indeed, both M-MDSCs and PMN-MDSCs specifically express *I1b*, *Arg2*, *Cd84*, and *Wfdc17* genes, thus highlighting a common immunosuppression-associated signature. Intriguingly, CD45^+^CD11b^+^Gr1^+^CD84^hi^ cells co-express the lectin-type oxidised LDL receptor 1 (LOX-1) that was reported to be a restrictive PMN-MDSC marker [49], and overall, these cells demonstrate a robust *in vitro* immunosuppression compared to CD45^+^CD11b^+^Gr1^+^CD84^lo^ cells [48]. It is now imperative to test this gene expression profile in other cancer settings to confirm CD84 as a specific MDSC marker.

In humans, the definition of MDSCs is more problematic, as several cell subsets have been identified [50] (Table 1). For example, an international consortium described six different MDSC phenotypes using multicolour staining based on conventional myeloid markers such as CD33, CD15, CD14, HLA-DR, and CD124 [51]. Generally, this cell identity complexity can be simplified by categorising human MDSCs into three main groups that include M-MDSCs (CD14^+^CD15^−^HLA-DR^lo/−^ cells), PMN-MDSCs (CD11b^+^CD14^−^CD15^+^CD66b^+^ low-density cells that are purified using a density gradient), and early-MDSCs (Lin^−^CD11b^+^CD34^+^CD33^+^CD117^+^HLA-DR^lo/−^ cells) that represent a small cell subset of myeloid precursors [41,52]. Although the main limitation of the current studies examining human MDSCs lies in the lack of standardisation of the experimental procedures of analysis [44,53], accumulating data have demonstrated that circulating MDSC frequency plays a critical role as an independent prognostic biomarker in various cancers [50,54,55,56] and can also serve as a useful predictive indicator of metastasis spread [38,57] and response to therapy [58,59,60,61]. The great amount of interest in MDSC research is also highlighted by recent data regarding the ability of MDSCs to facilitate immune functions in patients with COVID-19. Indeed, immunosuppressive monocytes that resemble M-MDSCs based on the expression of high levels of the arginase-1 (ARG1) enzyme were discovered during COVID-19 evolution [62], thus indicating a likely role for these cells in the progression to lymphopenia by actively inhibiting immune effectors. Therefore, MDSCs are key players not only in cancer-associated immune disorders but also in other immune-related pathologies such as viral and bacterial infections, autoimmune diseases, and inflammatory syndromes [41,63].

Distinctive cell-death-associated programs also facilitate M-MDSC and PMN-MDSC segregation in mice, a process that depends upon the myeloid cell leukemia (MCL)-1-mediated control of the intrinsic mitochondrial death pathway. In contrast, the anti-apoptotic molecule cellular FLICE (FADD-like IL-1β-converting enzyme)-inhibitory protein (c-FLIP) that functions as an important modulator of caspase-8 is crucial for the development of M-MDSCs [64]. By acting as moonlighting proteins, both c-FLIP and its viral isoform (v-FLIP) can activate the transcription of several immunosuppression-associated genes (i.e., *Il10*, *Il6*, *Cd38*, *Cd274*, and *Cd273*), in part through the activation of nuclear factor kappa-light-chain-enhancer of activated B cells (NF-κB) activation, and this does not affect the conversion of neutrophils into PMN-MDSCs [65,66]. Interestingly, c-FLIP^+^CD14^+^ monocytes isolated from patients with pancreatic ductal adenocarcinoma (PDAC) expressed high levels of CD38 and programmed death-ligand 1 (PD-L1), and an increase in c-FLIP^+^PD-L1^+^CD14^+^ cell number in combination with high levels of serum IL-6 has been identified as a negative independent prognostic factor for both overall survival and disease free survival (DFS) [65]. In agreement with these results, we demonstrated that in both SARS-CoV-2-infected hACE2 transgenic mice and autopsy samples from the lungs of patients with COVID-19, c-FLIP is overexpressed in myeloid cells, and c-FLIP-expressing human monocytes display immunosuppressive functions and release high amounts of pro-inflammatory cytokines [67]. Collectively, these data highlight FLIP as a key mediator for reprogramming monocytes into immune regulatory elements in both mice and humans.

Tumour-released soluble factors prompt imbalanced myelopoiesis that ultimately supports MDSC generation. Among these factors, granulocyte colony-stimulating factor (G-CSF), granulocyte macrophage colony-stimulating factor (GM-CSF), and macrophage colony-stimulating factor (M-CSF) play an essential role in controlling proliferation, maturation, and survival of myeloid cells [47,68], while pro-inflammatory cytokines and tumour-derived vesicles globally affect the normal route of haematopoietic stem cells (HSC) and myeloid precursor differentiation [31,69]. Consequently, the bone marrow (BM) in tumour-bearing hosts is chronically exposed to mediators that can inhibit the expression and function of canonical transcription factors involved in the myeloid differentiation at steady state condition (i.e., CCAAT/enhancer binding protein α (c/EBPα)), thus skewing the activation of “emergency” haematopoiesis regulators such as c/EBPβ and IFN regulatory factor (IRF)8 and members of the signal transducer and activator of transcription (STAT) family such as STAT3, STAT1, and STAT6 [70,71,72,73]. Among these regulators, c/EBPβ has emerged as an essential “master” regulator of MDSC expansion and immunosuppressive activity. Seminal work by Marigo et al. revealed that selective c/EBPβ deletion in myeloid lineage cells was sufficient to arrest MDSC genesis and completely abrogate their immunoregulatory properties by reducing ARG1 and inducible nitric oxide synthase (iNOS/NOS2) expression and activity, all of which are crucial components of the MDSC immunosuppressive machinery [71]. MDSC development through c/EBPβ-guided myelopoiesis is partially governed by the myeloid-specific expression of retinoic-acid related orphan receptor (RORC1/RORγ) that can suppress negative regulators of myelopoiesis such as suppressor of cytokine signalling 3 (Socs3) and B-cell lymphoma 3 (Bcl3) [74]. Conversely, downregulation of IRF8 in haematopoietic progenitors promotes the expansion of PMN-MDSCs, thus highlighting a critical role of IRF8 in neutrophil generation during emergency haematopoiesis [75,76]. Interestingly, these data were substantiated by the identification of a unique cluster of neutrophil precursors that were expanded in tumour-bearing mice that expressed low levels of IRF8 [77]. STAT3 is a key regulator of MDSC biology, as its genetic ablation in tumour-bearing mice abrogates MDSC expansion [78]. STAT3 not only promotes MDSC survival and proliferation by inducing the expression of both myc and anti-apoptotic proteins such as B-cell lymphoma XL (BCL-XL) [79], but it also fuels MDSC immunosuppressive properties. Indeed, STAT3 triggers the production of reactive oxygen species (ROS) by phagocytic oxidase [80] and promotes ARG1 activity due to the ability of its phosphorylated isoform (p-STAT3) to bind to different sites on the *Arg1* promoter to favour its transcription [81]. Recently, we demonstrated that M-MDSCs isolated from peripheral blood of PDAC patients, which are defined as immunosuppressive monocytes, exhibited upregulation of genes involved in fatty acid and lipoprotein metabolism (*Cd36*, *Lypla1*, and *Cers5*), ATP metabolism (*Atp5f1c*, *Atp5mc2*, and *Sdhb*), glucose metabolism (*Pdk4* and *Gxlt1*), and amino acid metabolism (*Erich1*, *Gls*, *Ctsc*, *Arg1*, *Nat2*, *Ust*, and *Oxr1*) compared to the expression levels of these genes in non-immunosuppressive monocytes. Furthermore, this immunosuppressive subset exhibits a unique STAT3-dependent expression of ARG1, thus confirming the pivotal role of STAT3 in guiding MDSC functions [57]. Finally, STAT3 elicits the release of the pro-inflammatory proteins S100A8/A9, which inhibit dendritic cell (DC) differentiation, ultimately favouring MDSC migration and accumulation within the tumour site [82,83]. MDSC functional maturation is also guided by the activation of NF-κB, which functions as the master regulator of inflammation. Based on the hypothesis that MDSCs are evolutionarily developed to control inflammation associated with seeding of the gut microbiota in neonates, Liu et al. demonstrated that lactoferrin (LF) converts newborn myeloid cells to MDSCs via the low-density lipoprotein-receptor-related protein-2 (LRP2) receptor and NF-κB activation [84]. The NF-κB pathway is activated in MDSCs by several mediators, such as toll-like receptor (TLR) ligands, S100A proteins, IL-1β, and TNFα [40]. Notably, in mice MDSCs express TLR2, TLR3, TLR4, TLR5, and TLR7/8/9, while TLR2 and TLR7/8 have been found in humans [85]. Recently, the compromised translocation of NF-κB p50 protein was reported to nullify the release of protein acidic and rich in cysteine (SPARC), thus altering ROS-dependent MDSC-associated immunosuppression. Additionally, the abrogation of p50 translocation into the nucleus impairs the generation of immunosuppressive p50:p50 homodimers in favour of the p65:p50 inflammatory heterodimers [86]. The critical role of NF-κB p50 protein in driving MDSC differentiation has been demonstrated by the axis that exists between tumour-derived prostaglandin E2 (PGE2) and the p50-associated NO-mediated immunosuppressive function of MDSCs [87] and by the acquisition of immunosuppressive function by monocytes following the enhancement of nuclear p50 translocation by c-FLIP [65]. Finally, MDSCs are also influenced by environmental and metabolic switches such as nonderepressible-2 kinase (GCN2), which detects the scarcity of any amino acids and constitutes the evolutionarily conserved amino acid starvation response pathway. Indeed, GCN2 regulates MDSCs’ and macrophages’ phenotype and suppressive function by enhancing the translation of activating transcription factor 4 (ATF4). [88]. The regulation of MDSC function and differentiation in TME also involves hypoxia-inducible factor 1 (HIF-1)α [89] and adenosine monophosphate-activated protein kinase (AMPK), another key sensor of cellular energy metabolism [90].

M-MDSCs and PMN-MDSCs are not fully committed to specific cell populations. Indeed, in tumour-bearing mice but not in tumour-free mice, M-MDSCs can act as a source of PMN-MDSCs through a mechanism that involves the epigenetic downregulation of the retinoblastoma protein (Rb) by histone deacetylase enzymes [91]. Moreover, splenic M-MDSCs (identified as CD11b^+^Gr1^int^ cells) isolated from tumour-bearing mice were able to reproduce the broadest spectrum of myeloid cell differentiation when transferred into tumour-bearing or tumour-free hosts, thus suggesting that M-MDSCs comprise inflammatory monocytes with multipotent progenitor features [92]. PMN-MDSCs are typically short-lived cells, and based on this, MDSC conversion into other myeloid subsets is better described in the context of M-MDSCs, as they are characterised by high cell plasticity. In the TME, M-MDSCs undergo sequential phenotypical changes such as the downregulation of Ly6C and CCR2 markers and differentiation in TAMs. This cell transformation is primarily mediated by M-CSF [93]. However, M-MDSC conversion in TAMs is not a conventional myeloid cell differentiation and instead strictly depends upon both the anatomical location and the tumour stage [52,94]. For example, we recently discovered that the disabled homolog 2 mitogen-responsive phosphoprotein (DAB2) plays a critical role in the reprogramming of pro-tumour macrophages [95]. Even if *Dab2* upregulation is mediated by M-CSF, it is not sufficient for full gene induction based on the observations that physical interactions between myeloid cells and extracellular matrix proteins are essential to guide the acquisition of a functional pro-tumour phenotype, thus highlighting the crucial impact of matrix stiffness on driving macrophage polarisation by YAP/TAZ signalling [95]. Moreover, the hypoxic environment of TME promotes the downregulation of STAT3 activity through a CD45 protein tyrosine phosphatase-dependent mechanism, thus favouring their differentiation to tumour-promoting TAMs [96]. As M-MDSCs are actively recruited to primary and metastatic tumour sites by several chemokine-associated circuits that are controlled primarily by C-C motif chemokine ligand (CCL)2 and CCL5, several antibodies and small molecules targeting this cell recruitment axis are being studied in different clinical trials for use both as monotherapies or in combination with standard therapy [97,98].

## 3. Metabolic Mechanisms of Immune Suppression by MDSCs

MDSCs have been identified as major suppressor cells in the context of anti-tumour immunity. Indeed, MDSCs can directly suppress effector functions of both NK cells and T lymphocytes through multiple mechanisms, and this has been extensively reviewed previously [31,40]. By releasing immunosuppressive cytokines such as transforming growth factor (TGF)β and IL-10, MDSCs promote the differentiation of Tregs and pro-tumour macrophages, respectively [99,100]. These inhibitory mediators also induce the downregulation of the expression of the activating receptor natural killer group 2 member D (NHG2D) in NK cells, thus lowering cytotoxic activity [101]. Concurrently, MDSCs express several ligands for inhibitory T cell receptors, and through the function of a T cell immunoglobulin and mucin-domain containing-3 (Tim-3)/Galectin (Gal)-9 pathway, MDSCs promote T cell exhaustion and the arrest of Th1-based immune response [102,103]. Moreover, tumour-associated hypoxia favours the upregulation of PD-L1 expression at the MDSC cell surface [104], and this suppresses T cell activation by binding to the immune checkpoint receptor programmed death 1 (PD-1).

However, the most pervasive mechanism employed by MDSCs to inhibit tumour-fighting cells is the metabolic reprogramming of the TME. Through a complex network of cellular components such as enzymes, transporters, and transcriptional factors acting in glucose-, lipid-, and amino acid-related metabolism, MDSCs strongly affect the anti-tumour immune response to establish a local tolerance within the tumour.

### 3.1. Glucose Metabolism

The cell energy source in the TME depends primarily on glycolysis, even in the presence of abundant oxygen, and this phenomenon is described as the Warburg effect [105]. Indeed, cells that undergo malignant transformation exhibit enhanced glucose uptake and excessive lactate production to generate biosynthetic precursors for supporting cell growth and proliferation. Compared to oxidative phosphorylation (OXPHOS), glycolysis is less effective in producing adenosine triphosphate (ATP), as a single particle of glucose generates only two ATP molecules. However, immune cells choose aerobic glycolysis due to the faster completion rate of this process compared to that of OXPHOS, and additionally, this process does not involve mitochondrial biogenesis and generates crucial biosynthetic mediators for use in other cellular processes (i.e., cell growth and proliferation).

Metabolic reprogramming of tumour cells plays a central role in MDSC development. Through the expression of liver-enriched activator protein (LAP) of c/EBPβ, aerobic glycolysis in breast cancer cells regulates the production and secretion of G-CSF and GM-CSF, both of which trigger MDSC development and accumulation within the TME [106] (Figure 1).

Moreover, the enhanced aerobic glycolysis in Epstein–Barr virus (EBV)-associated nasopharyngeal carcinoma prompts tumour cells to produce high amounts of IL-1β and IL-6, both of which are responsible for MDSC functions [107]. Accordingly, stimulation with IL-6 and GM-CSF of BM cells isolated from naïve mice is sufficient to activate l-arginine (l-Arg) metabolising enzymes, as well as increase glucose uptake and glycolysis, promoting the acquisition of BM-derived MDSC (BM-MDSC) phenotype [108]. However, *in vitro* blocking of l-Arg catabolising enzymes indirectly regulates glucose uptake in MSC-1 cells (an immortalised cell line derived from primary MDSCs), thus highlighting a close interaction between glucose and amino acid metabolism [108]. Among the different enzymes involved in the glycolytic pathway, pyruvate kinase (PK) plays a critical role in cancer progression. PKM2 is a splice variant of PK and is overexpressed in several cancers [109]. This protein fuels inflammation-associated programs in immune cells, inducing STAT3 phosphorylation and the subsequent production of inflammatory cytokines [110]. Indeed, PKM2 upregulation and phosphorylation in activated macrophages induces the generation of an unconventional PKM2/Hif-1α complex that binds to the IL-1β promoter and regulates the expression of Hif-1α-dependent genes. Currently, this metabolic network has been exclusively described in macrophages, and therefore, the role of PKM2 in MDSCs deserves further investigation.

Different cell subsets in the TME contribute to glucose consumption, including that of immune cells. Normally, glucose transport across biological membranes occurs primarily through either the glucose transporter (GLUT) or members of the solute carrier 2A (SLC2A) family [111]. Recently, 18F-fluorodeoxyglucose (FDG) positron emission tomography (PET) imaging that allows for the quantification of glucose uptake revealed that tumour-infiltrating myeloid cells rather than cancer cells preferentially absorb glucose by
a GLUT-1-based mechanism [112]. This phenomenon is conserved in several cancer models, including both genetically inducible and transplanted models. For example, in the MC38 colon cancer setting, higher glucose uptake was detected in two major immune cell populations that included M-MDSCs and TAMs [112]. Indeed, the partitioning of nutrients can be influenced by the mammalian target of rapamycin complex 1 (mTORC1) pathway that is typically upregulated in tumour-infiltrating myeloid cells [113]. Interestingly, a comparison between the same types of immune cells derived from two different conditions (tumour-free vs. tumour-conditioned) demonstrated that MDSCs exhibit a marked downregulation of genes encoding for glycolysis-related enzymes and glucose transporters compared to that of monocytes [114]. Indeed, MDSCs repressed GLUT-1 surface expression and hexokinase activity, ultimately leading to the establishment of a dormant metabolic phenotype by methylglyoxal glycation activity [114]. As methylglyoxal readily reacts with l-Arg metabolising enzymes, MDSCs transfer methylglyoxal to T cells where they deplete the intracellular enzyme resources by generating methylglyoxal-derived glycation products such as argpyrimidine and hydroimidazolone to thereby cause T cell paralysis. Therefore, methylglyoxal that is generated from acetyl-CoA and glycine through semicarbazide-sensitive amine oxidase (SSAO) activity can be considered as a potential metabolite marker for MDSCs [114].

Lactate is the final product of pyruvate conversion during aerobic glycolysis. Indeed, glucose is oxidised to pyruvate, and lactate dehydrogenase (LDH) is reduced to lactate [115]. As excess levels of the reduced form of nicotinamide adenine dinucleotide (NADH) suppress glycolysis, the action of LDH catalyses the regeneration of nicotinamide adenine dinucleotide (NAD^+^) to maintain the cellular NADH/NAD^+^ ratio necessary to sustain glycolytic metabolism [116]. Nicotinamide phosphoribosyl transferase (NAMPT) is a crucial enzyme in the nicotinamide adenine dinucleotide (NAD) salvage pathway that converts nicotinamide to nicotinamide mononucleotide (NMN), a precursor of NAD [117]. Recently, NAMPT has been reported to orchestrate the redistribution of myeloid cells in tumour-bearing hosts via the NAMPT–NAD^+^–SIRT1 pathway that promotes the activation of c/EBPα and c/EBP-β-driven *G-CSF* and *G-CSFR* gene transcription [118]. Accordingly, pharmacological inhibition of NAMPT prevents MDSC mobilisation and enhances the anti-tumour effectiveness of immunotherapy (i.e., anti-PD-1) [118].

Lactate has been identified as a critical signalling molecule involved in the regulation of several myeloid cell programs, including cell epigenetic reprogramming and cell-to-cell communication [115,119,120]. Indeed, the genetic silencing of tumour LDH reduced the number of tumour-infiltrating MDSCs in both triple-negative breast cancer (TNBC) [106] and melanoma [121] mouse models with a subsequent activation of anti-tumour immunity, thus highlighting the critical role of lactate in dampening cancer immune surveillance. However, lactate is uncoupled from the activation of genetic signatures in polarised macrophages [120], suggesting that the induction of immunosuppressive genes such as *Arg1* is independent of the lactate-associated pathway. Excess lactate in the TME is associated with the overexpression of lactate transporters such as monocarboxylate transporters (MCTs). These transporters are bidirectional symporters that mediate lactate efflux that is essential for the hyper-glycolytic phenotype, and they also contribute to pH regulation of the microenvironment, thus supporting the acid-resistant phenotype. Among the 14 different family members, MCT1 (also known as SLC16a1) is involved in the process where lactate and pyruvate is uploaded by the c-Myc-signalling pathway, while MCT4 (also known as SLC16a3) is an efficient lactate exporter that is induced by hypoxia and redox signals [115]. MCT transporters are required for their expression on the surface of the cell membrane and their interaction with co-chaperone immunoglobulin-family single-membrane pass proteins such as CD147 (also known as OX-47). Indeed, CD147 silencing inhibits the expression and function of MCT1 and MCT4, thus favouring an increase in intracellular lactate concentration in cancer cells [122]. Additionally, two sodium-coupled cotransporters known as SMCT1 (SLC5A8) and SMCT2 (SLC5A12) mediate the cellular uptake of lactate [123]. Currently, the role of lactate transporters in MDSC biology is poorly characterised and requires further investigation. Indeed, these proteins are potential attractive targets for developing innovative anti-cancer therapeutic strategies by limiting MDSC functions and abrogating the immunosuppressive features of TME, and this could open a new frontier for cancer immunotherapy.

The limited availability of energetic sources within hypoxic tumour areas induces the accumulation of extracellular ATP and its metabolite adenosine (ADO). The hydrolysis of ATP into ADP and/or AMP depends upon the ecto-enzyme CD39 (ectonucleoside triphosphate diphosphohydrolase 1; encoded by *Entpd1*), while the phosphorylase CD73 (or 5-NT) is responsible for AMT to ADO hydrolysis [124]. Extracellular ATP is generally considered to be a prototypical danger signal, as it accumulates at inflammatory sites and within the tumour interstitium at concentrations that may reach as high as the hundred micromolar range. It is undetectable in normal tissues [124]. Therefore, CD39 and CD73 produce an adenosine-rich TME that is suitable for immune evasion and tumour survival [125] and this is exemplified by data obtained using CD73-deficient mice that are able to activate an increased anti-tumour immune response compared to that of their normal counterparts [126]. Indeed, the induction of a CD39 isoform (CD39L1) in hepatocellular carcinoma cells prevents the differentiation of MDSCs into other myeloid cells, thus promoting their maintenance and accumulation [127]. Through TGF-β/mTOR/HIF-1 signalling, high levels of CD39 and CD73 were detected in MDSCs undergoing metabolic reprogramming in tumour-bearing mice [128,129,130]). Additionally, TGF-β can directly regulate the differentiation of intratumoural MDSCs into terminally differentiated mononuclear cells (CD45^+^CD11b^+^CD11c^+^F4/80^+^MHC II^+^Gr-1^−^) through the upregulation of CD39/CD73 [131]. However, blocking HIF-1α leads to the downregulation of CD39/CD73 ectoenzymes and the arrest of MDSC suppressive functions after pharmacological treatment with metformin [132]. In agreement with these observations, a positive correlation between the frequency of CD39- and CD73-expressing MDSCs in the TME and tumour stage has been identified in a cohort of patients with non-small-cell lung cancer [130].

Additionally, membrane channels that regulate ATP release by adenosine deaminase (ADA) and by specific receptors for both ATP (P2XR/P2YR) and adenosine (A1R, A2AR, A2BR, and A3R) control the adenosinergic pathway [125,133,134]. The presence of P2X7R in both M-MDSCs and PMN-MDSC subsets was initially discovered in the spleen of neuroblastoma tumour-bearing mice. However, the plasma membrane localisation of P2X7R has been associated with functional activity only in M-MDSCs [135]. Interestingly, MDSCs display P2X7R signalling that is uncoupled with caspase-3 function, thus leading to survival in an ATP-rich environment [135]. Among the ADO receptors, A2BR is attractive for its capacity to trigger immunosuppression primarily in myeloid cells [131,136] and to favour MDSC accumulation within tumours, as highlighted in melanoma experimental settings [137]. Collectively, the inhibition of these targets may provide a potential new therapeutic approach to enhance cancer immunotherapy.

### 3.2. Amino Acid Metabolism

Elevated consumption of amino acids is one of the mechanisms through which tumours evade the immune response. Tumour-associated myeloid cells exploit metabolic pathways to deplete the microenvironment of essential nutrients for T cells and to generate molecules possessing immunomodulatory characteristics. Among all amino acids, tryptophan, arginine, glutamine, and cysteine play a crucial role in regulating the viability, migration, and activation of T cells (Figure 2). 

l-tryptophan (l-Trp) is an essential amino acid in mammals and is involved in the synthesis of proteins, kynurenines (Kyn), and indole, and also in the endogenous production of NAD^+^. The majority of l-Trp is converted into kyn by indoleamine 2,3-dioxygenase 1-2 (IDO1 and IDO2) and tryptophan-2,3-dioxygenase (TDO). Alterations in the levels of l-Trp and its metabolites are associated with several diseases, including neurological disorders, autoimmunity, and cancer [138]. Recently, 3-hydroxy-l-kynurenamine (3-HKA), a biogenic amine that is produced via a lateral branch of the IDO1 pathway in both DCs and endothelial cells during inflammatory conditions as well as by cancer cells, has been identified as a crucial player in inhibiting CD8^+^ T lymphocyte proliferation and naïve-to-effector T cell maturation by blocking the STAT1/NF-κΒ pathway [139]. Among the l-Trp-converting enzymes, IDO1 plays a central role in orchestrating immune responses [138,140,141]. Indeed, its expression and activity have been described in several cancer types in response to IFNγ, and both have been correlated with poor prognosis [142]. In particular, MDSCs express high levels of IDO1 and promote the degradation of l-Trp in the extracellular environment, thus inhibiting T-cell function and suppressing the immune response [143,144]. Moreover, MDSCs support the expansion of Tregs and prevent DC immunogenicity by activating the aryl hydrocarbon receptor (AhR) [145,146]. *Ido1* deficiency in mice with lung cancer was associated with impaired MDSC expansion and immunosuppressive function, and also with reduced tumour growth and metastasis formation [147].

Over the last few decades, a multitude of compounds have been developed as IDO1 selective inhibitors for cancer immunotherapy and have been used in combinatorial therapies [148,149]. However, a number of clinical trials were terminated due to the lack of reported benefits [150], thus suggesting the need to establish a novel strategy to block l-Trp catabolism. Interestingly, a novel inhibitor of IDO1 that blocks the binding of the haem-free conformation of the enzyme (apo-IDO1) has been identified (GSK5628) and was recently synthesised [151]. In comparison to traditional drugs that bind to the iron atom of the IDO1 heme cofactor (holo-IDO1), this compound guarantees a durable inhibitory effect. Furthermore, a new class of IDO1 inhibitors (compound 10 and 23) exhibiting a distinctive binding mode in an uncommon target site was reported to significantly reduce l-Kyn levels in the plasma of mice and to impair the immunosuppressive function of human PDAC-derived MDSC-like cells [152].

By depriving the cellular microenvironment of l-Trp, IDO1 activates a stress response pathway involving the GCN2 kinase that causes the exit from the cell cycle and promotes the downregulation of T-cell receptor ζ-chain (TCR ζ) in CD8^+^ T cells to prevent the conversion of Tregs into T helper type 17 (Th17) cells [153,154,155]. *In vivo* deletion of GCN2 in the myeloid compartment was associated with tumour growth restriction in different transplantable tumour models and with the phenotypic switch of TAMs and MDSCs towards a pro-inflammatory phenotype via the translational induction of ATF4 [88]. Considering that GCN2 activation is correlated with reduced overall survival in patients with melanoma, GCN2 is an appealing target for immune modulation therapy [88]. Finally, IDO1 is not restricted to the cells and can also be secreted into extracellular vesicles (EVs), including exosomes. In fact, glioblastoma cells produce EVs containing IDO1, thus suggesting that tumours can exploit both intracellular and extracellular IDO1 to generate an immunosuppressive microenvironment [156]. This further highlights the complexity of IDO1-mediated pathways in the TME.

Interestingly, IDO1 should be considered as a moonlighting protein [66]. Indeed, IDO1 also acts as a signal-transducing molecule and fine-tunes the immune response over the long term in a TGFβ-dependent manner [157,158]. This function depends upon the presence of two immunoreceptor tyrosine-based inhibitory motifs (ITIMs) on the small, non-catalytic domain of the enzyme and a conserved post-transcriptional regulatory site located at a YENM motif. Upon tyrosine phosphorylation in ITIMs and at the YEMN domains in DCs, IDO1 binds and activates Src homology 2 domain-containing protein (SHPs) and class IA phosphoinositide 3-kinase (PI3Ks), thus promoting the activation of the non-canonical pathway of NF-κB that results in the upregulation of *Ido1* and *Tgfb1* genes [157] and the relocalisation of IDO1 from the cytosol to the early endosome (EE) for signalling, respectively [159]. In contrast, pro-inflammatory IL-6 upregulates SOCS3, which then binds to ITIMs via the SH2 domain, thus leading to the ubiquitination and consequent degradation of IDO1 [160]. Therefore, a drug that blocks the signalling activity of IDO1 in combination with its catalytic site may improve IDO1-targeting immunotherapy. Recently, it was discovered that under anaerobic conditions IDO1 is an efficient nitrite reductase capable of generating NO [161]. This study provides new insights into the role of IDO1 as a novel enzyme source of NO in tumours.

Despite numerous studies regarding IDO1 and its function, we are not aware of the l-Trp cell-entry mechanism. Indeed, the molecular identity of the transporter remains elusive, and additional studies are required to develop a class of inhibitors that can abrogate not only the enzyme function but also l-Trp transportation [162,163].

l-Arginine (l-Arg) is a conditionally essential amino acid that is required by mammals under specific circumstances, such as the activation of immune responses [164]. l-Arg is metabolised from urea and l-ornithine (Orn) by ARG1 or ARG2 and from nitric oxide (NO) and l-citrulline by iNOS. Moreover, Orn is involved in the synthesis of polyamines (putrescine, sperimidine, and spermine) and proline. Many cancers are characterised by intense ARG activity that is associated with impaired T cell function. ARG1 is induced in tumour-infiltrating myeloid cells such as TAMs, MDSCs, and DCs through the secretion of tumour-derived factors and metabolites, including IL-4, IL-13, IL-6, M-CSF, GM-CSF, TGF-β, and cAMP [52,165]. ARG1 expression in myeloid cells is modulated by several transcription factors and nuclear receptors, such as peroxisome proliferator-activated receptors (PPARγ-δ), STAT6, STAT3, c/EBPβ, IRF8, c/EBP homologous protein (CHOP), PU.1, Kruppel-like factor 4 (KLF4), and activator protein 1 (AP-1) transcription factors [52]. ARG1 possesses a hierarchical negative function compared to that of iNOS in establishing an immunosuppressive TME, as iNOS-expressing cells such as tumour-infiltrating M1-polarised macrophages [166] and Tip-DCs (a DC subset that releases TNF and NO) [167] are actively involved in tumour debulking. Recently, we demonstrated that suppressive M-MDSCs are significantly more represented in patients with PDAC than they are in healthy donors and that they can be identified as circulating STAT3/ARG1-expressing monocytes that can be clearly distinguished from other circulating monocytes [57]. A peculiar characteristic of MDSCs is that they express both ARG1 and iNOS to mediate Arg catabolism and suppress the T cell response. Anti-inflammatory cytokines (IL-4, IL-13, and IL-10) secreted by PMN-MDSCs induce ARG1 activity that leads to downregulation of TCR ζ expression [168] and blockade of T cell cycle progression in the G0-G1 phase [169]. In contrast, pro-inflammatory cytokines (IFNγ, TNFα, and IL-1) in M-MDSCs support NO production [170]. Co-expression of l-Arg catabolising enzymes promotes the generation of RNS in the TME such as peroxynitrite (ONOO^−^), which can affect not only the structure of proteins but also modify the protein–protein interaction and function [170]. Notably, nitrogen-derived post-translational modifications (PTMs) of CCL2 drastically affect tumour homing of CD8^+^ T lymphocytes, while in contrast, nitrated/nitrosylated CCL2 does not lose the ability to engage MDSCs in the TME [171].

Notably, in the presence of TGF-β the l-Arg and l-Trp metabolic pathways are connected in a tangled relationship. TGF-β upregulates both ARG1 and IDO1 in DCs and causes a more rapid induction of ARG1 than of IDO1. ARG-expressing DCs produce Orn and spermidine, and these in turn promote the phosphorylation of IDO1 and its long-term signalling activity [149,172]. MDSCs that express high levels of ARG1 and are an abundant source of TGF-β can condition DCs to trigger IDO1 expression via ARG1 metabolites, thus potentiating their immunoregulatory phenotype. Therefore, the simultaneous inhibition of ARG1 and IDO1 and/or TGFβ signalling could constitute a compelling anti-cancer therapeutic strategy.

An alternative approach to reduce l-Arg metabolism by MDSCs is to target amino acid carriers. MDSCs uptake l-Arg through cationic amino acid transporters (CATs) such as CAT1 and CAT2 that are constitutively expressed in several tissues or induced *in vitro* by inflammatory cytokines (such as IFN-γ, IL-13, and GM-CSF), respectively [173]. In a prostate cancer mouse model, CAT2 expression has been demonstrated to be upregulated in MDSCs that are recruited to the tumour site, while *Cat2* deletion *in vivo* increased T cell expansion and affected both tumour growth and the suppressive ability of MDSCs [173]. These data can be useful to better investigate CATs in the context of building innovative MDSC targeting approaches.

Recently, two distinct immune cell populations expressing ARG1 have been discovered within the TME of a fibrosarcoma tumour mouse model. ARG1-expressing cells were classified as TAMs or monocytic regulatory cells with immunosuppressive function (mirroring M-MDSCs) and were characterised by the expression of triggering receptors expressed on myeloid cells 2 (TREM2) [174]. *Trem2* knockout mice exhibited decreased recruitment of myeloid regulatory cells in tumours that was associated with increased T and NK cell functionality and reduced tumour growth [174], thus highlighting the potential therapeutic effect of targeting *Trem2* signalling to reduce immune suppression activity.

l-glutamine (l-Gln) is the most abundant amino acid in humans, and it is involved in the synthesis of glutathione (GSH), amino acids, and nucleotides, in protein glycosylation, and in the production of intermediate α-ketoglutarate for the TCA cycle [175]. Although l-Gln is considered to be a nonessential amino acid, it is necessary for cell proliferation and tumour expansion [176]. Considering that glutaminolysis also sustains MDSC maturation and suppression activity [177], l-Gln blockade can be regarded as an appealing therapeutic target (NCT04471415). The novel l-Gln antagonist 6-diazo-5-oxo-l-norleucine (DON) has been demonstrated to significantly decrease tumour growth in various transplantable cancer models, to enhance T cell function, and to establish immunological memory upon tumour re-challenge [178]. Moreover, the same compound is able to inhibit the generation and accumulation of MDSCs in the tumour and metastasis sites in different mouse models via reduction of c/EBPβ expression and in turn levels of G-CSF, one of the factors responsible for MDSC recruitment to the TME [179]. Despite no differences in the total number of TAMs in tumours, the l-Gln antagonist promotes the acquisition of a proinflammatory phenotype. This phenomenon is not limited to TAMs and is also extended to TAM precursors such as MDSCs [179]. In a pioneering study based on the MSC-1 cell line, an *in vitro* model of MDSCs, the absence of l-Gln has been reported to affect iNOS activity without perturbing ARG1 function [177]. Interestingly, blockade of glutamine metabolism negatively regulates IDO expression in both tumour and myeloid-derived cells, thus supporting anti-tumour immunity [179]. Collectively, inhibiting l-Gln transporters or glutaminolysis enzymes in MDSCs may offer a promising means by which to recover the anti-tumour immune response.

l-cysteine (l-Cys) is an essential amino acid that is required for T-cell activation and that functions as a building block for proteins. Normally, cells generate cysteine from methionine via cystathionase [180]. Alternatively, cells can import cysteine through their surface cystine/glutamate antiporter SLC7A11 (also commonly known as xCT) [181]. Recently, SLC7A11 was identified as a key molecular determinant involved in protection from oxidative stress and the control of Treg cell proliferation under normal and pathological conditions [182]. Considering that T cells lack cystathionase, they only take up Cys produced by leukocytes in the TME such as macrophages and DCs. Therefore, the anti-tumour function of T cells is inhibited by MDSCs that limit the pool of Cys in the extracellular environment due to their ability to import cysteine with approximately the same kinetics as macrophages and DCs. However, in contrast to these antigen-presenting cell subsets, MDSCs do not export cysteine, as they do not express the alanine-serinecysteine (ASC) transporter [181]. Considering the overlapping activity of these transporters, it became challenging to discriminate between their upregulation and cysteine uptake in cancer. However, treatment with an engineered cyst(e)inase has been demonstrated to promote cell cycle arrest and death in cancer cells in both breast and prostate preclinical models due to depletion of intracellular GSH that protects the cells from oxidative stress and is essential for their survival and proliferation [183]. Finally, loss of intracellular cysteine contributes to the induction of ferroptosis, an iron-dependent cell death process characterised by the accumulation of iron-dependent lethal lipid peroxides (LPO) that can induce membrane rupture and ultimately lead to cell death [184,185]. MDSCs are protected from ferroptosis by the expression of high levels of system xCT and neutral ceramidase *N*-acylsphingosine amidohydrolase (ASAH2) [186]. Moreover, MDSCs in the TME accumulate high levels of arachidonate (AA)-esterified triglycerides (AA-TAGs), oxidised AA-TAGs, and PGE2, which is a major oxidative metabolite of free AA and limits the peroxidation of AA, thus acquiring resistance to ferroptosis [187]. Moreover, myeloid cells in the TME can release interleukin-4-induced-1 (IL4i1), a secreted amino acid oxidase that mediates the production of indole-3-pyruvate (I3P) from tryptophan, to thus block ferroptosis and support tumour growth [188]. Precise mechanisms in the regulation of ferroptosis in cancers must be further explored, even though the complexity of the biological system and the difficulty in clinical translation bring challenges and opportunities to the further development of ferroptosis-based cancer therapies.

### 3.3. Lipid Metabolism

MDSCs’ commitment in the TME includes crucial cellular metabolic reprogramming to sustain high energy demands and their suppressive and tumourigenic functions [189]. For this reason, tumour-infiltrating MDSCs boost the oxidation of fatty acids (FAO), thus shifting their dominant source of energy from glycolysis toward FAO. Consequently, mitochondrial mass expansion, upregulation of key FAO enzymes, and increased oxygen consumption rates follow. FAO in turn produces acetyl-CoA that participates in the tricarboxylic acid cycle and oxidative phosphorylation, ultimately producing the required molecules of ATP [190]. Acetyl-CoA is the primary building block for the biosynthesis of fatty acids, cholesterol, and other complex lipids in the cytosol (Figure 3).

This MDSC metabolic adaptation has been reported in both mouse models and human cancer [190] and can be achieved by upregulating exogenous fatty acid (FA) uptake through the expression of FA transport proteins (FATPs) and the scavenger receptor FA translocase CD36 [191]. The overexpression of CD36 requires STAT3 and STAT5 signalling activation, both of which are triggered by tumour-released G-CSF and GM-CSF. Genetic deletion of CD36 could significantly hamper lipid uptake, accumulation, and oxidation, thus causing inhibition of tumour growth and tumour-infiltrating MDSC immunosuppressive function. Notably, STAT3 and STAT5 may be potential targets to control metabolic and functional MDSC polarisation. Indeed, inhibitors such as FLLL32 or pimozide that interfere with JAK2/STAT3 and STAT5, respectively, not only induce intracellular neutral lipid reduction but also prevent the induction of ARG1 and iNOS [191].

Due to the uncontrolled growth of malignant cells, the TME is a hostile milieu characterised by hypoxia, nutrient deprivation, acidosis, and an elevated production of ROS and RNS that in turn induces disruption of calcium and lipid homeostasis, ultimately resulting in endoplasmic reticulum (ER) stress. Three intraluminal protein sensors that include inositol-requiring enzyme 1 (IRE-1)α, protein kinase RNA (PXR)-like ER kinase (PERK), and ATF6 can dissociate from the ER-chaperone BiP to sense this homeostatic imbalance and initiate the specialised signalling cascade activating the unfolded protein response (UPR). UPR activation typically restores ER homeostasis and promotes adaptation to diverse insults in the TME, thus ensuring cell survival and autophagy or leading to programmed cell death [49,192,193,194,195]. PERK phosphorylates eukaryotic protein synthesis initiation factor alpha (eIF2α) that inhibits the initiation of *de novo* protein synthesis and reduces protein loading in the ER. Consecutively, eIF2α triggers ATF4 expression and its downstream targets, including CHOP, which has already been shown to control TME-infiltrating MDSC survival and immunosuppression activity [192,196]. Conversely, IRE-1α undergoes oligomerisation, thus activating its RNase domain which cleaves X-box binding protein-1 (*XBP-1*) mRNA. Indeed, sXBP-1 spliced fragments activate several genes encoding ER chaperones and foldases, lipid biosynthesis enzymes, and ER-associated degradation mediators, where the latter is also activated by the ATF6 UPR branch [197]. It is not surprising that UPR sensors have been proposed as potential therapeutic targets to overcome MDSC-mediated T cell dysfunction. Seminal work by Mohamed et al. revealed that PERK deletion in MDSCs restrained the signalling pathway of NRF2, a master regulator of antioxidant genes, to promote the production of type I IFN by sensing the accumulation of cytosolic mtDNA through STING [198]. Indeed, myeloid cell-conditional *Eif2ak3^−/−^* mice (*Eif2ak3*^fl/fl^; *Lyz2*-cre) that possess abrogated PERK exhibit functional reprogramming of M-MDSCs and PMN-MDSCs and also overexpression of anti-tumour cytokines and tumour growth arrest [198]. Alicea-Torres et al. demonstrated that the immunosuppressive functions of MDSCs depend upon the p38 kinase-associated downregulation of IFNAR1 that inactivates the IFN1 pathway [199]. Notably, short-term treatment with low-dose PERK inhibitors establishes an enforced anti-tumour immune response, thus boosting the anti-PD-L1 effectiveness but without inducing side effects [198].

Free fatty acids, triacylglycerol-carrying lipoproteins, very-low-density lipoproteins (VLDL), and low-density lipoproteins (LDL) in the TME not only accumulate but also undergo oxidation, and their oxidised counterparts can affect MDSC functions and survival [200]. Among the several recent studies examining ER stress inducers, oxLDL is certainly one of the major areas of interest. Indeed, Condamine et al. introduced a new concept according to which the conversion of neutrophils to PMN-MDSCs depends directly upon the expression of LOX1 that is a key receptor for oxLDL that can trigger immunosuppressive functions in myeloid cells [49]. As a result, LOX1 was proposed as a marker of a sub-population of immunosuppressive ER-stressed PMN-MDSCs (4–7% of total neutrophils) in patients with cancer rather than in mice. As recently reviewed elsewhere [201], LOX1 may provide an excellent target to restore immune function in a broad range of patients with cancer, particularly those with glioblastoma (GBM) where LOX1^+^ PMN-MDSC was demonstrated to play a prominent role in driving cancer development [202]. More studies in different patient cohorts are needed to confirm LOX1 as a suitable cell marker to identify and target MDSCs.

Excessive amounts of cholesterol under oxidative stress conditions promote the generation of oxysterols. Oxysterols and other cholesterol derivatives are commonly sensed by Liver-X (LXRα/β) nuclear receptors that regulate cholesterol and lipid metabolism by inducing the transcription of genes involved in reverse cholesterol transport. LXRβ and one of its transcriptional targets, apoliprotein E (ApoE), have both been thoroughly studied in the last decade for their potential role in pro-metastatic invasion and tumour neo-angiogenesis [203]. Tavazoie et al. demonstrated that LXR activation induced by an LXRβ agonist treatment elicits strong anti-tumour effects in both tumour-bearing mice and patients with cancer. Across a broad array of murine cancers such as lung, ovarian, renal cell, triple-negative breast, and colon cancer, and also in melanoma and glioblastoma models, LXRβ treatment promotes partial or complete tumour regression and a reduction in both tumour-infiltrating and circulating PMN-MDSCs and M-MDSCs [204].

Adeshakin and colleagues previously demonstrated the role of polyunsaturated fatty acids (PUFA) in MDSC expansion and immunosuppression activity in tumours via JAK/STAT3 pathway activation [205], and they recently proposed a new role for FATP2 in regulating MDSC immunosuppressive activity through lipid-accumulation-induced ROS. Indeed, the small molecule inhibitor lipofermata was identified as a new approach to overcome resistance to anti-PD-L1 therapies that rely entirely upon the GM-CSF/STAT3/STAT5/FATP2 axis [206]. Of note, lipofermata upregulates the expression of NRF2 both in tumour-bearing mice and patients, thus suggesting that FATP2 regulates immunosuppressive MDSC functions through an anti-oxidative signalling pathway. Interestingly, FATP2 is expressed in both M-MDSCs and PMN-MDSCs generated from the *in vitro* differentiation of BM cells and can also be isolated from the spleens or BMs of LLC-tumour-bearing mice [206], thus contradicting the previous knowledge that FATP2 is mainly expressed in PMN-MDSCs [207]. Collectively, these considerations allow us to assume that the alteration of MDSC lipid metabolism is a tumour-specific event that requires contextualised investigations.

FATPs replenish MDSCs from a broad range of lipids, including arachidonic acid that is essential for the production of PGE2 and the immunosuppressive activity of PMN-MDSCs [207]. However, PGE2 could even be considered as an epigenetic ‘hacker’ of M-MDSCs functions, as it drives the accumulation of nuclear NF-κB p50 protein in M-MDSCs purified from the spleens or BM of fibrosarcoma-tumour-bearing mice and also from human circulating M-MDSCs isolated from patients with colorectal cancer [87]. Indeed, nuclear NF-κB p50 accumulation epigenetically reprograms the response of monocytes, thus leading to iNOS upregulation and establishing a pro-tumour phenotype. Therefore, targeting the PGE2/p50/NO axis may provide a valid strategy to improve anticancer immunotherapy.

## 4. MDSC Targeting as a Therapeutic Approach in Cancer

### 4.1. Current MDSC-Targeting Approaches

Controlling MDSC accumulation and immunosuppression is an attractive strategy to reduce tumour progression. Indeed, several MDSC-targeting approaches have proven to be effective and well-tolerated therapies in diverse preclinical settings [208]. Consequently, an increasing number of clinical trials are ongoing to evaluate the safety and efficacy of MDSC inhibition alone or in combination with other cancer therapies (radiotherapy, chemotherapy, surgery, or immunotherapy, Table 2). Currently, therapeutic strategies aimed at eliminating MDSCs and/or abrogating their pro-tumour activities can be divided into four distinctive classes according to their ability to promote tumour reduction, and this has been previously reviewed [208,209].
MDSC depletion. The use of conventional chemotherapy drugs such as gemcitabine, 5-fluoruracil, paclitaxel, and doxorubicin demonstrated the ability of these compounds to restore the anti-tumour immune response by controlling the accumulation of MDSCs in both tumour-bearing mice [92] and patients with cancer [60,61]. More recently, “peptibodies” consisting of S100A9-derived peptides conjugated to antibody Fc fragments have exhibited the ability to regulate MDSC expansion in tumour-bearing mice [210], and the use of antibodies recognising CD33 (a surface marker expressed by human MDSCs) conjugated with ozogamicin was able to affect myeloid cells accumulation in patients with acute promyelocytic leukaemia [211]. Therefore, this myeloid-cell-targeting approach may be exploited in solid cancer settings to block MDSCs. Ongoing clinical trials are also evaluating the therapeutic effects of modulating growth factors such as G-CSF (NCT02961257) or M-CSF (NCT02880371, NCT02554812, and NCT02777710, Table 2) to improve the effectiveness of cancer therapy by preventing MDSC expansion and accumulation.Inhibition of MDSC recruitment to the tumour site. Interrupting the chemokine and chemokine receptor axis is a rational strategy to prevent MDSC trafficking. Indeed, therapeutic blockade of CCL2-CCR2 interaction using CCL2 neutralising antibodies or a CCR2 antagonist as well as blocking the heterotypic CXCL5-CXCR2 signalling circuit produces promising results in several preclinical cancer models by affecting MDSC accumulation and restoring anti-tumour immunity [212,213,214]. Interestingly, low doses of methyltransferase and histone deacetylase inhibitors affect MDSC accumulation in lung premetastatic niches by affecting CCR2- and CXCR2-dependent pathways [215]. Currently, ongoing clinical trials are testing the therapeutic impact of targeting the CC or CXC axis to block MDSC trafficking as a single therapy (NCT01349036) or in combination with either chemotherapy (NCT02370238) or ICI-based therapy (NCT03177187 and NCT03161431, Table 2).Blocking MDSC differentiation into mature suppressive cells. One promising therapeutic approach depends upon the conversion of MDSCs into non-suppressive elements of the TME. Early studies demonstrated that all *trans*-retinoic acid (ATRA), a derivative of vitamin A, possesses the potential to force the differentiation of MDSCs into mature granulocytes and upregulate glutathione synthase (GSS), one of the enzymes required for glutathione synthesis [216]. As the enforced expression of IRF8, either directly or indirectly, may be a potential strategy to favour MDSC reprogramming, several strategies are able to sustain IRF8-dependent signatures such as STAT3 inhibition [217]. Sorafenib, a multikinase inhibitor, was able to reverse the immunosuppressive cytokine profile in tumour-infiltrating myeloid cells, thus promoting them to elicit a more robust anti-tumour immune response [218]. A phase II clinical trial is currently investigating the impact of ATRA in combination with ICI-based therapy to decrease the immunosuppressive activity of MDSCs in patients with melanoma compared to that in patients treated with immunotherapy alone (NCT02403778).Inhibition of MDSC-suppressive activity. Impeding MDSC immunosuppression by targeting key enzymes, signal transduction molecules, or transcription factors involved in the MDSC metabolic network is likely the most effective strategy to enforce cancer immunotherapy. Here, we report only some example strategies based on pivotal interfering metabolism-associated molecules that have not been discussed in detail in the previous sections.

As tumour-reprogrammed M-MDSCs displayed a significantly increased expression of both mTOR and glycolysis-associated genes, including *Glut1*, *Hk2*, *Gpi*, *Tpi*, *Eno1*, *Pkm2*, *Ldha*, and *MCT4*, mTOR targeting by rapamycin not only decreased glycolysis but also inhibited the expression of immunosuppression-associated targets such as ARG1 and PD-L1. These results support the use of rapamycin for cancer therapy and the possibility of combining this drug with conventional therapies to enhance anti-tumour effects [219]. The use of dichloroacetate (DCA), a pyruvate dehydrogenase kinase (PDK) inhibitor, significantly reduced lactate release, STAT3 activation, IDO1 upregulation, and MDSC infiltration in preclinical models [220]. Similarly, targeting PI3K gamma that is a crucial signalling molecule required for myeloid cell accumulation in inflammation reverts immunosuppression in the TME by impairing MDSC migration in both pancreatic and skin cancers [221,222]. The combination of eganelisb (a potent inhibitor of PI3K) with nivolumab is currently under evaluation in a phase I trial in patients with advanced solid tumours (i.e., breast cancer, mesothelioma, melanoma, and non-small lung cancer) (NCT02637531), as well as in combination with atezolizumab in a phase II trial in patients with triple-negative breast cancer or renal cell carcinoma (NCT03961698).

STAT3 plays an important role in the activation of essential components of l-Arg-dependent MDSC metabolism and immunosuppression ([223] and reference therein). STAT3 inhibitors such as AZD9150 antisense oligonucleotide block STAT3 mRNA translation [224,225], and sunitinib [226] downregulates STAT3 phosphorylation and reduces *Arg1* expression, thus ultimately contrasting ARG1-dependent MDSC suppressive activity. AZD9150 is now being tested in a phase I/Ib clinical trial in patients with advanced/metastatic hepatocellular carcinoma (NCT01839604). Although targeting of STAT3 has been extensively investigated for decades, it is traditionally regarded as an undruggable target due to its high toxicity and the poor bioavailability of developed drugs. However, some FDA-approved compounds such as pyrimethamine and celecoxib were identified as STAT3 inhibitors through drug-repositioning screening [227,228], thus suggesting potential applications of these drugs for cancer therapy. Alternatively, indirect inhibitors of STAT3 have been developed by targeting the upstream or downstream components of the STAT3 signalling pathway. Of these, JAK- and IL-6-associated targeting have been approved by the FDA for cancer therapy and are currently being tested in ongoing clinical trials (NCT01112397, NCT01423058, NCT03427866, NCT01594723, NCT02055781, NCT03315026, and NCT02997956). Recently, we demonstrated that the JAK inhibitor baricitinib that is normally used to treat rheumatoid arthritis reduces systemic inflammation in patients with COVID-19 by reducing inflammatory cytokine levels, thus promoting lymphocyte restoration and modifying the immune-suppressive features of myeloid cells [229].

l-Arg metabolism generates polyamines that play an important role in tumour growth promotion [230]. Inhibition of polyamine biosynthesis by targeting ornithine decarboxylase (ODC) and the concomitant blockade of polyamine import in tumour cells has been proposed as a therapeutic strategy. α-difluoromethylornithine (DFMO), an irreversible inhibitor of ODC activity that is currently used in clinics for its anti-carcinogenesis activity, was demonstrated in combination with the Trimer polyamine transport inhibitor to significantly reduce tumour growth in comparison to that observed in response to treatments alone by decreasing the number of tumour-infiltrating MDSCs and by improving CD8^+^ T cell activation in two different tumour mouse models [231]. Additionally, this combinatorial therapy was reported to significantly enhance the anti-tumour efficacy of PD-1 blockade in tumour-bearing mice that are resistant to ICI monotherapy [232]. Several preclinical studies have suggested the effectiveness of the ARG1-targeting approach in controlling tumour growth [149]. Among them, the novel ARG-inhibitor CB-1158 limits tumour progression in different preclinical settings [233]. An ongoing phase II clinical trial is currently testing the safety profile of CB-1158 in advanced solid tumours in a combination therapy ICI (NCT02903914). Phosphodiesterase-5 (PDE-5) inhibitors inhibit MDSC functions by downregulating both iNOS and ARG1 expression and activities. In preclinical mouse models, administration of PDE-5 inhibitors such as sildenafil and tadalafil reactivates T- and NK-mediated anti-tumour immunity [234,235]. Completed clinical trials examining PDE-5 inhibitors (NCT02544880, EudraCT-No: 2011-003273-28) have revealed enhanced intratumoural T cell activity and improved patient outcomes in head and neck squamous cell carcinoma and metastatic melanoma [236,237].

Several compounds targeting IDO-1 catalytic activity have demonstrated anti-tumour effects in different preclinical settings. More than 20 ongoing clinical trials are investigating the ability of IDO-1 to enhance the therapeutic effect of standard anti-cancer therapies [149]. Interestingly, a completed phase I clinical study (NCT02048709) of the orally active IDO1 inhibitor navoximod (also referred to as GDC-0919 or NLG-919) demonstrated the ability of this drug to stabilise tumour progression in 36% of treated patients as a single agent [238]. Moreover, another completed phase I clinical trial (NCT02471846) of navoximod combined with atezolizumab in patients with advanced or metastatic solid tumours revealed that the treated patients achieved either a partial response (9%) or a complete response (11%) [239]. Therefore, IDO inhibitors in combination with ICI-based immunotherapies or standard treatments deserve thorough investigations in cancer, and more clinical studies with immune monitoring are required to evaluate the impact of IDO-1-targeting on modulating the immunological landscape of TME.

Although it is a promising therapeutic approach to control cancer progression, limited clinical research data regarding glutaminase inhibition are currently available. Recently, a phase I/IIa multi-centre, non-randomised, multi-cohort, open-label study was conducted to evaluate the therapeutic impact of sirpiglenastat (a glutamine antagonist known as DRP-104) in patients with advanced solid tumours (NCT04471415). Analysis of immune modifications such as the abrogation of immunosuppression in TME in patients treated with l-glutamine targeting will be required to provide new strategies for cancer immunotherapy.

Finally, targeting systemic PGE2 with cyclooxygenase 2 (COX-2) inhibitors has been demonstrated to block MDSC accumulation in preclinical cancer models [240,241]. A number of clinical trials examining COX-2 inhibitor combination therapies are currently ongoing in patients with colon carcinoma, breast cancer, and squamous cell carcinoma of the head and neck (NCT03026140, NCT04188119, NCT04348747, and NCT03245489). However, selective COX-2 inhibitors induce critical side effects such as gastric ulcers or increased risk of cardiovascular events. Thus, COX-2 inhibitors should be carefully administered and should exclude patients affected by specific contraindications. Unfortunately, there are currently no clinical trials exploring COX-2 inhibition in MDSCs.

Collectively, preclinical and clinical studies support the employment of MDSC-targeting strategies to enhance host immune response, improving the efficacy of current cancer immunotherapy protocols.

### 4.2. New Perspectives in MDSC-Targeting Approaches

Increasing technological progress coupled with knowledge progression in the fields of basic immunology, molecular biology, and biochemistry have been a “game changer” in the oncoimmunology research field. The discovery of new immunosuppressive switches such as c-EBPβ [71] or c-FLIP [65] has been recently supported by innovative approaches to differentially target (molecular rather than protein) myeloid cells (specifically MDSCs) in many pathological contexts. The introduction of new specific and powerful gene editing tools such as lentivirus and clustered regularly interspaced short palindromic repeats (CRISPR)-associated proteins to modify immune cells represents a large step forward in this field, which has advanced from the preclinical research stage to clinical trial evaluation [242] in a timely manner. CRISPR technology was originally developed to knock-out or knock-in specific genes in target cells by employing a target-specific 20 nucleotide RNA sequence (crRNA) coupled with a small non-coding trans-activating crRNA (tracrRNA). The binding of the complex to the target DNA sequence and the association of nucleases to a protospacer-adjacent motif (PAM) near that sequence guides the generation of a single- or double-stranded DNA cleavage, thus triggering either non-homologous end-joining (NHEJ) or homology-directed repair (HDR) processes. Since its discovery, the CRISPR system has been constantly developed to suit new applications, including the ability to act at the genetic, epigenetic, and RNA levels [243]. Gene editing tools have recently been employed to modify the TME from a pro-tumour to an anti-tumour milieu. Enforced expression of miR-142-3p in BM-MDSCs impairs their immunosuppressive abilities [244]. Myeloid cells engineered to produce IL-12 can alter the immune-suppressive metastatic environment and sustain T cell activation, and these processes result in a lower metastatic burden and improved overall survival in preclinical models of sarcoma and pancreatic cancer [245]. Although gene editing tools targeting immunosuppressive myeloid cells are currently primarily exploited for basic and translational immunology studies, recent advances in the development of tools that specifically deliver drugs into target cells *in vivo* and the employment of CRISPR-based transient regulation of gene expression [243] will hopefully extend these applications to the clinic. Nanoparticle-based platforms allow for specific targeting *in vivo*. Gemcitabine-loaded PEGylated lipid nanocapsules target M-MDSCs in both the periphery and tumour, thus relieving immunosuppression and improving T-cell-based immunotherapy [246]. Similarly, gemcitabine-loaded lipid-coated calcium phosphate nanoparticles reshape the TME by depleting MDSCs, skewing macrophage polarisation toward an M1 phenotype, and promoting anti-tumour immunity in melanoma [247]. Notably, modifications of the nanoparticle surface charge dictate the subset of immune cells that will be targeted, where neutral particles are primarily engulfed by monocytes and M-MDSCs and negative particles are predominantly captured by macrophages in tumour samples of patients with glioma [248]. Alternatively, dual-pH-sensitive vesicles have been designed and tested to directly transport immunomodulatory drugs into the TME [249]. These premises establish a solid basis for CRISPR-loaded nanoparticle-mediated approaches to relieve MDSC-derived immunosuppression, and these could soon be translated into new cancer immunotherapies. According to this, another revolutionary approach to hijack MDSC function and metabolism relies upon a recently developed technology termed proteolysis-targeting chimaeras (PROTAC) that exploits the ubiquitin proteasome system to modulate target proteins at the post-translational level [250]. PROTAC employs heterobifunctional degraders based on E3 ligase ligands attached to a linker and a targeting module that allow for interaction with the target protein, thus supporting its ectopic ubiquitination and consequent degradation in the proteasome [250]. This protein degradation toolbox is rapidly expanding and currently includes more than 1600 publicly available heterobifunctional degraders that recognise approximately 100 targets [251]. Indeed, PROTAC possesses many advantages over the inhibitors and gene editing tools that are already available on the market. First, efficient target coverage (compared to that of inhibitors) and tertiary structure-dependent interactions ensure highly specific targeting. Moreover, transient binding to the target is sufficient to trigger its degradation. Finally, post-translational modifiers do not modify the cellular genome, and they can therefore reach the clinic faster. For example, lenalidomide that was recently approved for patients with multiple myeloma [252] recruits the E3 ligase cereblon to target the Ikaros transcription factor (IKZF1). However, it has been recently observed that IKZF1 plays a pivotal role in promoting macrophages’ skew towards a pro-tumour phenotype, and lenalidomide interrupts this switch, supporting their repolarisation towards a tumouricidal one [253]. New “degraders” that can target tumour cells and support myeloid cell “re-education” are enlisted in this development pipeline. Kymera therapeutics last year began two phase I clinical trials examining KT-474 (targeting IRAK4) and KT-333 (targeting STAT3) candidates to treat autoimmune diseases and patients with cancer, respectively [254]. These recent advances hold new promise for targeting MDSC-dependent immunosuppression, preventing MDSC-supported tumour spreading, and supplying solid soil to design innovative immunotherapy approaches for patients with cancer.

## 5. Conclusions

MDSCs are the cornerstone of immunosuppression in the TME and protect cancer cells from immune attack. Indeed, elevated levels of circulating and tumour-infiltrating MDSCs are negative prognostic biomarkers for ICI, thus highlighting MDSCs as the major obstacle blocking the effectiveness of cancer immunotherapy [59]. Therefore, it is necessary to fully characterise the complex network of interactions within the TME that sustain, activate, and protect MDSCs to develop new therapeutic approaches to eliminate or reprogram MDSCs. Following the encouraging results obtained in mouse models using MDSC-targeting approaches (i.e., histone-deacetylase inhibitor or IPI-145, an inhibitor of phosphatidylinositol-4,5-bisphosphate 3-kinase (PI3Kδ and PI3Kγ isoforms)) in combination with ICI-based therapy [255,256], a number of early-phase clinical trials are attempting to circumvent MDSC-dependent T cell inhibition and improve ICI outcomes in patients with cancer [257]. In the next few years, the possibility of modulating MDSCs by tuning their metabolic pathways may provide unexpected immunostimulatory cues favouring immune-mediated anticancer effects. Nevertheless, several leading questions regarding the development of metabolic targeting of MDSCs remain unanswered. First, the identity of the key nodes of the complex metabolic network underlying the MDSC capacity to rapidly activate their immunosuppressive functions in response to environmental perturbations of the TME must be determined. Second, we must characterise the specific metabolic determinants that can discriminate among MDSC subgroups and between M-MDSCs and tumour-infiltrating TAMs and between PMN-MDSCs and tumour-infiltrating neutrophils. Third, we must understand how the metabolic reprogramming of the TME influences MDSC biology. Finally, the best targeting approach to alter MDSC-associated phenotypes and co-opted immunosuppressive functions must be identified.

Research conducted over the next few years will pave the way for elucidating these questions and integrating myeloid-targeting approaches into the clinical care of patients with cancer.

## Figures and Tables

**Figure 1 cells-10-02700-f001:**
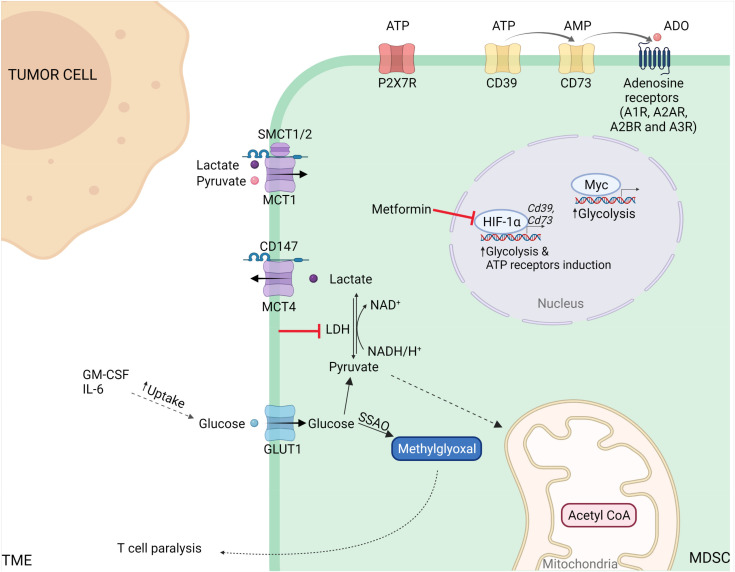
The effect of tumor microenvironment on the glucose metabolism of MDSCs. Cytokines (i.e., GM-CSF, IL-6) secreted by cancer cells boost both glucose uptake and MDSC glycolysis, leading to an increased production of lactate, that is transported across plasma membrane through monocarboxylate transporters (MCTs). Additionally, hypoxia regulates the glycolytic pathway of MDSC by hypoxia-inducible factor (HIF-)1α, that enhance the immunosuppressive features and the CD39/CD73 ecto-enzymes spread on MDSC membranes. Furthermore, the TME is rich in extracellular adenosine triphosphate (ATP) and its metabolites (ADP, adenosine diphosphate; ADO, adenosine), which can either stimulate P2X7R receptor or be degraded to ADO by the consecutive action of CD39 and CD73. NAD or NADH, nicotinamide adenine dinucleotide; SSAO, semicarbazide-sensitive amine oxidase; GLUT1, glucose transporter 1; SMCT, sodium-coupled monocarboxylate transporter; LDH, lactate dehydrogenase.

**Figure 2 cells-10-02700-f002:**
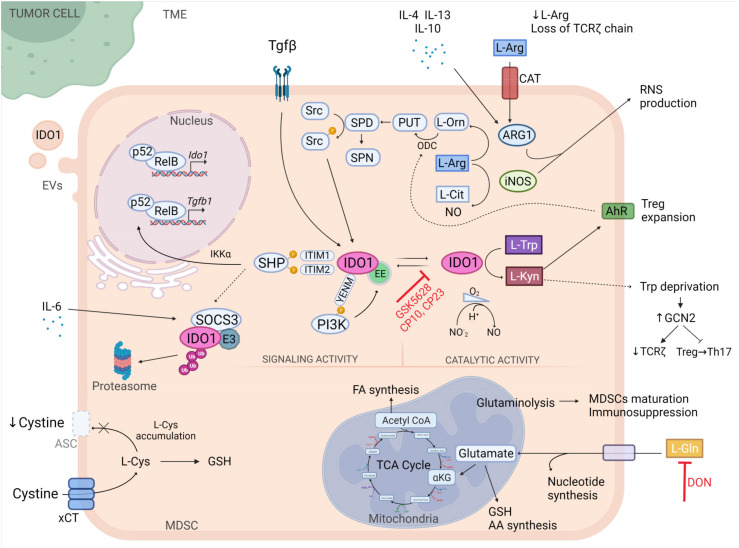
Metabolic conversion of amino acids contributes to the immunosuppressive role of MDSCs. l-arginine (l-Arg) is converted into urea and l-ornithine (l-Orn) by arginase-1 (Arg1), and into l-Citrulline (l-Cit) and nitric oxide (NO) by inducible nitric oxide synthase (iNOS). l-Orn is further metabolized by ornithine decarboxylase (ODC) into polyamine (PUT, putrescine; SPD, spermidine; SPN, spermine). SPD promotes the phosphorylation of indoleamine 2,3-dioxygenase 1 (IDO1) through the activation of the Src kinase. Once phosphorylated, IDO1 recruits Src homology region 2 domain-containing phosphatase (SHPs) and phosphoinositide 3-kinase (PI3K), triggering the non-canonical NF-kb pathway and anchoring IDO1 to the early endosome (EE), respectively. SHP binding induces the phosphorylation of Inhibitory-κB Kinase α (IKKα) and nuclear translocation of p52-RelB complex, which stimulate the transcription of genes encoding for IDO1 and transforming growth factor beta 1 (TGFβ), creating a positive feedback loop. On the contrary, interleukin (IL)-6 presence upregulates suppressor of cytokine signaling 3 (SOCS3), which in turn recruits E3 ubiquitin ligase (E3) and drives IDO1 to proteasome degradation. Moreover, IDO1 degrades l-tryptophan (l-Trp) and produces l-kynurenine (l-Kyn), an agonist of the aryl hydrocarbon receptor (AhR), leading to *Ido1* upregulation and regulatory T cells (Treg) expansion. Trp deprivation by IDO1 activates nonderepressible-2 kinase (GCN2), causing the downregulation of T-cell receptor ζ-chain and inhibition of the conversion of Treg in T helper type 17 (Th17) cells. IDO1 also acts as nitrite reductase to generate NO in anaerobic conditions and can be secreted in extracellular vesicles (EVs). l-Cysteine (l-Cys) is imported by Cystine/glutamate antiporter xCT (xCT) in the cytosol of MDSCs, where it accumulates due to the lack of alanine-serinecysteine (ASC) transporter. l-Glutamine (l-Gln) participates to nucleotide, glutathione (GSH), amino acid, fatty acid (FA) synthesis, and glutaminolysis, supporting MDSC maturation and immunosuppression activity. TME, tumor-microenvironment; CAT, cationic amino acid transporters; RNS, reactive nitrogen species; ITIM1-2, immunoreceptor tyrosine-based inhibitory motifs 1-2; TCA, tricarboxylic acid cycle; CP, compound.

**Figure 3 cells-10-02700-f003:**
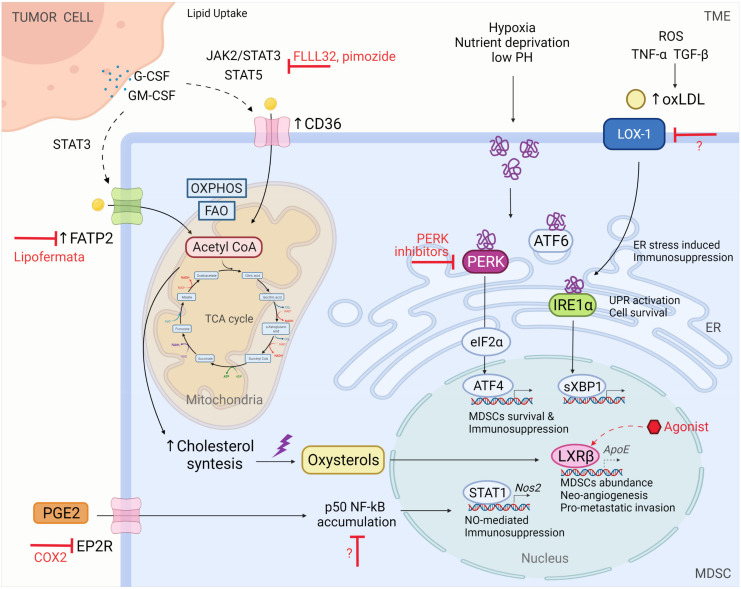
Lipid homeostasis disruption mediates MDSC immunosuppressive function. To sustain their suppressive and tumorigenic functions, MDSCs boost fatty acids oxidation (FAO) and oxidative phosphorylation (OXPHOS). FAO produces Acetyl-CoA, which is essential for ATP and endogenous lipid synthesis. Meanwhile, exogenous fatty acid (FA) uptake occurs thanks to the upregulation of FA transporters such as CD36 and fatty acid transport proteins (FATPs) on the cell surface. Among the broad set of lipids by which MDSCs are replenished, there is arachidonic acid, relevant for prostaglandin E2 (PGE2) production. PGE2 drives nuclear accumulation of p50 protein of NF-κB, necessary for NO production and NO-mediated immunosuppression. On the other side, the hostile nature of the tumor-microenvironment (TME) induces calcium and lipid homeostasis disruption leading to ER stress and, as a consequence, unfolded protein response (UPR) activation. Oxidative stress conditions also mean lipid oxidation: while oxidized low-density lipoproteins (ox-LDLs) from the TME mediate ER stress-induced immunosuppression involving LDL receptor-1 (LOX-1) receptor, oxysterols through the liver-X nuclear receptor (LXR)/ apoliprotein E (ApoE) axis mediate MDSCs survival and abundance, regulating cholesterol and lipid metabolism. FA, fatty acids; FATP2, fatty acid transport protein 2; ROS, reactive oxygen species; TNF-α, tumour necrosis factor α; TGF-β, transforming growth factor β; oxLDL, oxidized low-density lipoprotein (LDL); Nos2, nitric oxide synthase 2; LXRβ, liver-X β nuclear receptor; PERK, RNA (PXR)-like ER kinase; IRE1α, inositol-requiring enzyme 1; eIF2α, eukaryotic initiation factor 2 α; sXBP1, spliced X-box binding protein-1; EP2R, PGE2 receptor; COX2, cyclooxygenase 2.

**Table 1 cells-10-02700-t001:** Phenotype of human MDSC subset.

	M-MDSCs	PMN-MDSCs	Early-MDSCs
**CD45**	+	+	+
**CD11b**	+	+	+
**CD33**	+	+	+
**HLA-DR**	low/−	−	−
**IL4Rα/CD124**	+	+	ND
**CD34**	+	ND	+
**CD84**	+	+	ND
**DR5**	+	+	ND
**CD14**	+	-	ND
**CD15**	−	+	ND
**Lin (CD3, CD19, CD20, CD56)**	ND	ND	−
**CXCR1**	+	low	ND
**CD66b**	−	+	ND
**CD117**	ND	ND	+
**CD38**	+	+	ND
**PD-L1/2**	+	ND	ND
**LOX1**	ND	+	ND
**FATP2**	ND	+	ND
**FSC**	low	low	low
**SSC**	low	high	high

**Table 2 cells-10-02700-t002:** Current MDSC targeting clinical trials.

NCT	Drug	Target	Combination Therapy	Phase	Effect	Tumour Type
NCT02961257	Granulocyte colony-stimulating factor (G-CSF)	Growth factor modulation	Prednisone, cabazitaxel	III	MDSC depletion	Prostate cancer, metastatic
NCT02880371	ARRY-382 (cFMS tyrosine kinase inhibitor)	CSF1R inhibitor	Pembrolizumab	II	MDSC depletion	Advanced solid tumours
NCT02554812	PD 0360324 (M-CSF mAb)	CSF1R inhibitor	Utomilumab, PF-04518600, avelumab, CMP-001	II	MDSC depletion	Locally advanced or metastatic solid tumours
NCT02777710	Pexidartinib (CSF-1R TKI)	CSF1R inhibitor	Durvalumab	I	MDSC depletion	Colorectal cancer, pancreatic cancer, metastatic cancer, and advanced cancer
NCT01349036	Pexidartinib hydrochloride (PLX3397)	binds to and inhibits phosphorylation of KIT, CSF1R and FLT3		II	MDSC trafficking	Recurrent glioblastoma
NCT02370238	Reparixin	CXCR1/2 inhibitor	Paclitaxel	II	MDSC trafficking	Metastatic triple-negative breast cancer
NCT03177187	AZD5069	CXCR2 antagonist	Enzalutamide	II	MDSC trafficking	Metastatic castration-resistant prostate cancer
NCT03161431	SX-682	CXCR1/2 inhibitor	Pembrolizumab	I	MDSC recruitment	Metastatic melanoma
NCT02403778	ATRA	Retinoic acid receptor inhibitor	Ipilimumab	II	Inhibition of MDSC immunosuppression	Advanced melanoma
NCT02637531	IPI-549 (Eganelisib)	PI3K-gamma inhibitor	Nivolumab	I	MDSC reprogramming and decreased immunosuppression	Advanced solid tumours
NCT03961698	IPI-549 (Eganelisib)	PI3K-gamma inhibitor	Atezolizumab, nab-paclitaxel, bevacizumab	II	Inhibition of MDSC immunosuppression	Triple-negative breast cancer and renal cell carcinoma
NCT01839604	AZD9150	STAT3 antisense oligonucleotide		I	Inhibition of MDSC immunosuppression	Advanced/metastatic hepatocellular carcinoma
NCT01112397	AZD1480	JAK1/2 inhibitor		I	Inhibition of MDSC immunosuppression	Advanced solid tumours
NCT01423058	Momelotinib (CYT387)	JAK1/2 inhibitor		I/II	Inhibition of MDSC immunosuppression	Myeloproliferative neoplasms
NCT03427866	Ruxolitinib	JAK1/2 inhibitor		II	Inhibition of MDSC immunosuppression	Myelofibrosis
NCT01594723	LY2784544	JAK2 inhibitor		II	Inhibition of MDSC immunosuppression	Myeloproliferative neoplasms
NCT02055781	Pacritinib	JAK inhibitor		III	Inhibition of MDSC immunosuppression	Myeloproliferative neoplasms
NCT03315026	Siltuximab	IL-6 inhibitor		II	Inhibition of MDSC immunosuppression	Multiple myeloma or systemic amyloidosis
NCT02997956	Tocilizumab	IL-6 inhibitor		II	Inhibition of MDSC immunosuppression	Hepatocellular carcinoma
NCT02903914	CB-1158 (INCB001158)	Arginase inhibitor	Pembrolizumab	I/II	Inhibition of MDSC immunosuppression	Advanced/metastatic solid tumors
NCT02544880	Tadalafil	PDE-5 inhibitor	Pembrolizumab	I	Inhibition of MDSC functions	Head and neck squamous cell carcinoma, head and neck cancer
EudraCT-No:2011-003273-28	Tadalafil	PDE-5 inhibitor				Metastatic melanoma
NCT02048709	Navoximod (GDC-0919)	IDO1 inhibitor		I		Solid tumour
NCT02471846	Navoximod (GDC-0919)	IDO1 inhibitor	Atezolizumab	I		Advanced/metastatic solid tumours
NCT04471415	Sirpiglenastat (DRP-104)	Glutamin antagonist	Atezolizumab	I/II		Advanced solid tumours
NCT03026140	Celecoxib	COX-2 inhibitor	Nivolumab, ipilimumab	II	Inhibition of MDSC accumulation	Colon carcinoma
NCT04188119	Aspirin	COX-2 inhibitor	Avelumab, lansoprazole	II	Inhibition of MDSC accumulation	Triple negative breast cancer
NCT04348747	Celecoxib	COX-2 inhibitor	Rintatolimod	II	Inhibition of MDSC accumulation	Metastatic breast cancer
NCT03245489	Acetylsalicylic acid	COX-2 inhibitor	Pembrolizumab, clopidogrel	I	Inhibition of MDSC accumulation	Head and neck squamous cell carcinoma

## Data Availability

Not applicable.

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
