# Peer review of "A Complex Metabolic Network Confers Immunosuppressive Functions to Myeloid-Derived Suppressor Cells (MDSCs) within the Tumour Microenvironment"

_cells, 2021, doi:10.3390/cells10102700_

Round 1

Reviewer 1 Report

This is an excellent review summarizing recent findings in the field of MDSCs metabolism. The authors describe how metabolic alterations in MDSCs are associated with tumor progression and response to therapy. They further summarize and critically analyze different therapeutic strategies for targeting MDSCs. 

Author Response

We thank the Reviewer for the beautiful comments about our manuscript.

Reviewer 2 Report

Hofer, Sario, Musiu et al. present a review of metabolic processes involved in MDSC differentiation and functions in pathologies, more specifically in the tumor microenvironment. It is rather complete and well written. It summarizes the main aspects and more on the subject.

Please find my comments below:

line 93-99: The authors list some factors that MDSC use to limit anti-tumoral response, but is this function limited to these factors? are these the main factors?

Line 116-117: Fate mapping of MDSCs is indeed critical. A few articles came out this year with single cell dataset specifically on myeloid cells. Can the authors precise if any data on MDSCs, differenciation and funciton came out from these datasets?

Line 137: Can the authors precise which lineage (Lin) it is?

Table 1: Suggestion: the table may be reorganized a bit.  Here we see individual markers for each MDSCs subsets. Maybe a list of markers and positivity and negativity in each subset would allow a better comparison.

line 159: Can the authors give example of immunosuppression-associated genes?

line 176-177: What about anti-inflammatory cytokines such as TGFb? Do they participate to myeloid precusor differenciation?

line 208-209: This is very interesting, how do these molecules affect the cycles. Positively or negatively? 
Also, any comparison with macrophages? since they are tissue resident and might also express CD14?

line 221-224: If NfKB on MDSC is activated by TLRs, can the authors list which TLRs are expressed by MDSCs, so that we may understand to which stimuli they may respond to. 

line 236: TGCN2 regulates MDSCs and macrophages differenciation towards which profile? or from what type of precursors? Also is it present when mMDSCs differenciate to TAMs?

line 299: please precise which cellular processes are incvolved.

line 307: please describe how BM-MDSCs can be generated, which factors are used.

line 367: Can the authors precise to which immunotherapy the response is enhanced?

line 404: are there any human data available too?

line 411: terminally differenciated mononuclear cell is a bit vague. Can the authors give examples?

line 558-564: Can the authors describe if a direct comparison between DCs and MDSCs was assessed according to these parameters?

line 569: Can the authors give example or the main inflammatory cytokines involved in CAT1 and CAT2B process.

line 589-604: There is an glutamine antagonist currently being tested in phase I clinical trial for solid tumor. NCT04471415. Can the authors mention it?

In general:  Can the authors list the clinical trials in a table.

I think that this review will be suitable for publication after responding to these comments.

Author Response

We thank the reviewer for all the productive comments. Please see the attachment in which we address all raised questions.
